# *Rhodopsin* targeted transcriptional silencing by DNA-binding

Salvatore Botta[1], Elena Marrocco[1], Nicola de Prisco[1], Fabiola Curion[1], Mario Renda[1], Martina Sofia[1], Mariangela Lupo[1], Annamaria Carissimo[1], Maria Laura Bacci[2], Carlo Gesualdo[3], Settimio Rossi[3], Francesca Simonelli[3], Enrico Maria Surace[1,4]*

[1]Telethon Institute of Genetics and Medicine, Napoli, Italy; [2]Department of Veterinary Medical Sciences, University of Bologna, Bologna, Italy; [3]Multidisciplinary Department of Medical, Surgical and Dental Sciences, Eye Clinic, Second University of Naples, Naples, Italy; [4]Department of Translational Medicine, University of Naples Federico II, Naples, Italy

**Abstract** Transcription factors (TFs) operate by the combined activity of their DNA-binding domains (DBDs) and effector domains (EDs) enabling the coordination of gene expression on a genomic scale. Here we show that in vivo delivery of an engineered DNA-binding protein uncoupled from the repressor domain can produce efficient and gene-specific transcriptional silencing. To interfere with RHODOPSIN (*RHO*) gain-of-function mutations we engineered the ZF6-DNA-binding protein (ZF6-DB) that targets 20 base pairs (bp) of a *RHO* cis-regulatory element (CRE) and demonstrate *Rho* specific transcriptional silencing upon adeno-associated viral (AAV) vector-mediated expression in photoreceptors. The data show that the 20 bp-long genomic DNA sequence is necessary for *RHO* expression and that photoreceptor delivery of the corresponding cognate synthetic trans-acting factor ZF6-DB without the intrinsic transcriptional repression properties of the canonical ED blocks *Rho* expression with negligible genome-wide transcript perturbations. The data support DNA-binding-mediated silencing as a novel mode to treat gain-of-function mutations.

*For correspondence: surace@tigem.it

**Competing interests:** The authors declare that no competing interests exist.

## Introduction

Transcription factors (TFs) operate by entangling their DNA-binding and transcriptional activation or repression functions (*Ptashne, 2014*). However, in eukaryotes TF DNA binding and effector activities are typically structurally modular (*Brent, 1985*) consisting of a DNA-binding domain (DBD) controlling the TF topology on genomic targets and an effector domain (ED) (*Brent, 1985*; *Kadonaga, 2004*) that recruits co-activator or co-repressor complexes (*Malik and Roeder, 2010*; *Perissi et al., 2010*) resulting in either transcriptional activation or repression of gene regulatory networks (GRNs) (*Neph et al., 2012*). Engineered TFs mimic the design of natural TFs (*Pavletich and Pabo, 1991*; *Beerli and Barbas, 2002*). To generate target specificity the DBD module is engineered to recognize unique genome sites (*Beerli and Barbas, 2002*), whereas the transcriptional activation or repressor properties are conferred by the selection of the ED (*Konermann et al., 2013*). To silence gain-of-function mutations, while studying the features of genomic DNA-TF interactions, here we investigated the hypothesis that engineered DNA-binding proteins without canonical ED activity possess transcriptional repression properties. As a transcriptional repression target we selected the G-protein-coupled Receptor Rhodopsin (RHO) gene whose gain-of-function mutations are those most commonly associated with autosomal dominant retinitis pigmentosa (adRP), an incurable form of blindness (*Dryja et al., 1990*).

**eLife digest** Proteins called transcription factors bind to sections of DNA known as regulatory elements to activate or deactivate nearby genes. In animals, transcription factors typically have two sections: a "DNA-binding domain" that attaches to DNA, and an "effector domain" that is responsible for interacting with other proteins to regulate the gene's expression.

Rhodopsin is a gene that encodes the instructions needed to make a light-sensitive protein in the eyes of humans and other animals. Botta et al. have now used this gene as an example to investigate whether proteins that contain a DNA-binding domain – but not an effector domain – can repress gene expression.

The experiments show that only a small section of the regulatory elements in the human Rhodopsin gene is actually required for the gene to be expressed. Botta et al. designed an artificial protein – referred to as ZF6-DB – that is able to bind to this section of DNA. The binding of ZF6-DB to this short DNA section was sufficient to switch off a Rhodopsin gene in living pig cells, and, unlike conventional transcription factors, seemed to have minimal impact other genes.

Next, Botta et al. used a virus to insert both the gene that encodes ZF6-DB and a normal copy of Rhodopsin into pigs. In these animals, ZF6-DB switched off the existing copy of Rhodopsin, but not the inserted copy so the cells produced a working form of the light-sensitive protein. Further experiments were carried out in mice that have both a faulty version and a normal copy of the Rhodopsin gene. ZF6-DB switched off the faulty Rhodopsin gene, which allowed the normal Rhodopsin gene to work without any interference from the faulty copy.

Mutations in Rhodopsin can cause an eye disease that leads to severe loss of vision in humans. These new findings could now guide future efforts to develop treatments for people with this condition. It will also be important to investigate how ZF6-DB binds to the regulatory elements in the Rhodopsin gene and whether a similar strategy could be used to alter the expression of other genes.

We generated a DNA-binding protein targeted to a cis-regulatory element (CRE) of the human proximal RHO promoter region by deconstructing an engineered TF (synthetic) composed of a DBD (ZF6-DNA-binding protein, ZF6-DB) and the ED (Kruppel-associated box, KRAB repressor domain, KRAB), which we have shown to be effective in repressing specifically the human RHO transgene carried in an adRP mouse model (Mussolino et al., 2011a). The deletion of the ED resulted in a protein, ZF6-DB targeting 20 base pairs of genomic CRE, here named ZF6-cis, found at -84 bp to -65 bp from the transcription start site (TSS) of the human RHO gene (Figure 1a; Mitton et al., 2000). Genomic ZF6-cis is without apparent photoreceptor-specific endogenous transcription factor-binding sites (TFBS; Figure 1a), as reported (Kwasnieski et al., 2012). To study the CRE features of ZF6-cis that ZF6-DB would interfere with upon binding in the absence of KRAB-mediated co-repressor recruitment, we deleted the 20 bp genomic ZF6-cis sequence and assessed its function by eGFP reporter assay (Kwasnieski et al., 2012) in living porcine retina by AAV delivery. The 776 bp-long RHO promoter fragment carrying the ZF6-cis deletion upstream of the eGFP reporter gene (AAV-RHO-cis-del-EGFP), after delivery to the porcine retinal photoreceptor, showed loss of eGFP expression compared to the control vector (AAV-RHO-EGFP) (Figure 1b,c). This suggests that ZF6-cis CRE is necessary for RHO expression (at least for the 776 bp region used in the assay) and that binding of the synthetic ZF6-DB trans-acting counterpart of ZF6-cis, may indeed repress RHO transcription.

Chromatin immunoprecipitation (ChIP) experiments to evaluate binding to the ZF6-cis target genomic sequence showed occupancy by the DNA-binding protein ZF6-DB (Figure 1—figure supplement 1b). To evaluate whether ZF6-DB represses transcription of the RHO gene in a physiological genomic context, we used the porcine retina (Mussolino et al., 2011b), which shares 19 out of 20 DNA bp with the human genomic ZF6-cis sequence (Figure 1—figure supplement 1a). Subretinal delivery of a low AAV8 vector dose ($1\times10^{10}$ genome copies; gc) of ZF6-DB (AAV8-CMV-ZF6-DB) resulted in a 45% decrease of porcine Rho transcript levels at 15 days post-injection (Figure 1d). Immunofluorescence analysis showed depletion of Rho protein and consequent collapse of the rod outer segments in ZF6-DB positive cells (Figure 1e,f). Despite the lack of detectable Rho expression

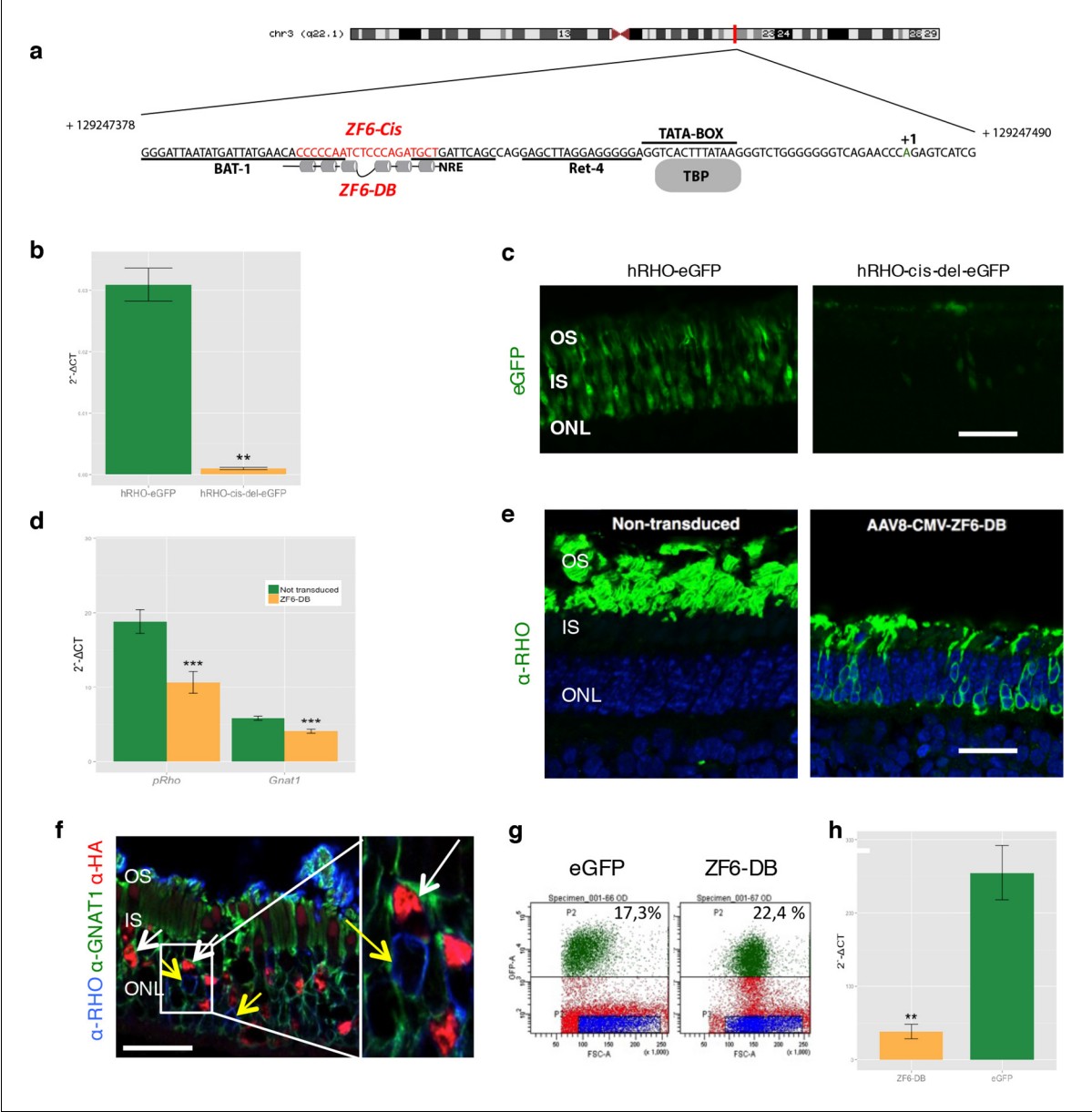

**Figure 1.** Delivery of ZF6-DB DNA-binding synthetic trans-acting factor targeted to a 20 bp of *RHO* cis-acting regulatory element (CRE) dramatically reduces *Rho* expression in photoreceptors. (**a**) Schematic representation of the chromosomal location of the *RHO* locus and its proximal promoter elements indicating the transcription start site (in green, +1) and the location of ZF6-DB binding site (in red, ZF6-Cis) and ZF6-DB (based on Mitton *et Al.,* 12); BAT1, Bovine A/T-rich sequence1; NRE, NRL response element; TBP, TATA box binding protein. (**b**) qReal Time PCR of mRNA levels ($2^{-\Delta CT}$) on the adult porcine retina 15 days after vector delivery of either AAV8-hRHO-eGFP (n=2) or AAV8-hRHO-cis-del-eGFP (n=2) subretinally administered at a dose of $1\times10^{10}$, showed that AAV8-hRHO-cis-del-eGFP resulted in decreased transduction (about fifty fold) compared with hRHO. (**c**) Histology confirmed the decrease of eGFP expression in hRHO-cis-del-eGFP injected retina compared with the retina injected with hRHO-eGFP. Scale bar, 50 μm. (**d**) qReal Time PCR of mRNA levels ($2^{-\Delta CT}$) of adult porcine retina injected subretinally with AAV8-CMV-ZF6-DB (n=6) at a vector dose of $1\times10^{10}$ genomes copies (gc) compared with non-transduced area (n=7) of the same eye 15 days after vector delivery, resulted in robust transcriptional repression of the *Rho* transcript. *pRHO*, porcine *Rhodopsin*; *Gnat1*, Guanine Nucleotide Binding Protein1. (**e**) *Rho* Immunofluorescence (green) histological confocal analysis of AAV8-CMV-ZF6-DB treated porcine retina compared with non-transduced area. Scale bar, 100 um. The treatment with ZF6-DB determined collapse of the outer-segment (OS) with apparent retention of nuclei (stained with DAPI) in the outer nuclear layer (ONL). (**f**) Immunofluorescence triple co-localization staining of porcine retina shown in (**b**) with Rho (blue), rod specific protein Gnat1 (green) and HA (ZF6-DB, red) antibodies. White arrows indicate co-localization of both HA-tag-ZF6-DB and *Gnat1* rods depleted of *Rho*, whereas yellow arrows showed residual *Rho* and *Gnat1* positive cells lacking ZF6-DB. A magnification of the triple staining (box) is highlighted. Scale bar, 100 μm. OS, outer segment; IS, inner segment; ONL, outer nuclear layer; INL, inner nuclear layer. (**g**) Representative fluorescence-activated cell sorting (FACS) of porcine retina 15 days after injections of either AAV8-GNAT1-eGFP (dose $1\times10^{12}$ gc) or co-injection with both AAV8-GNAT1-eGFP and AAV8-CMV-ZF6-DB (dose of eGFP, $1\times10^{12}$

*Figure 1 continued on next page*

Figure 1 continued

gc; ZF6-DB dose 5x10^10 gc). eGFP positive sorted cells (AAV8-GNAT1-eGFP) corresponded to 17,3% of the analysed population (left panel; P2 area, green dots), whereas, 22,4% of eGFP positive cells in the retina that received both vectors (AAV8-GNAT1-eGFP and AAV8-CMV-ZF6-DB; right panel; P2 area, green dots). (h) qReal Time PCR on sorted rods treated with AAV8-GNAT1-eGFP (n=3) and AAV8-CMV-ZF6-DB (n=3) showed a repression of about 85% of total rhodopsin when compared with rods treated with eGFP (mRNA levels: $2^{-\Delta CT}$). Error bars, means +/- s.e.m. n =; *p<0.05, **p<0.01, ***p<0.001; two-tailed Student's t test.

The following figure supplement is available for figure 1:

**Figure supplement 1.** Chromatin Immunoprecipitation (ChIP) of ZF6-DB and ZF6-KRAB.

in most of the transduced rods, rows of photoreceptor nuclei were preserved from degeneration at this time point (*Figure 1e*). To further evaluate the extent of *Rho* silencing in rod photoreceptors by ZF6-DB, we performed FACS analysis on eGFP-labelled rod cells. Rod cells were isolated from porcine retina that had received a subretinal injection of an AAV vector encoding eGFP under the control of a rod-specific promoter (human *Guanine Nucleotide Binding Protein1, GNAT1* promoter elements (*Lee et al., 2010*); AAV8-GNAT1-eGFP; dose 1x10^12 gc) with or without the vector encoding ZF6-DB (5x10^10 gc). Fifteen days after injection, the retina were disaggregated and FACS-sorted. The retina co-transduced with eGFP and ZF6-DB vector showed virtually a 'somatic knock-out' of *Rho* expression (~85% decrease of Rho transcript levels; *Figure 1g,h*).

To evaluate genome–wide transcriptional specificity, we analyzed the porcine retinal transcriptome by RNA sequencing (RNA-Seq) from retina harvested 15 days after subretinal injection of the AAV8 vector encoding ZF6-DB (*Figure 2*). For comparison we used the engineered TF with the ED, KRAB (AAV8-CMV-ZF6-KRAB). The low vector doses delivered to the porcine retina (1x10^10 gc) resulted in about twenty-fold lower expression levels of the ZF6-DB and ZF6-KRAB transgenes compared to *Crx* and *Nrl*, two retina-specific TFs (*Swaroop et al., 2010*) (*Figure 2a*). Of note, despite these low expression levels, we observed robust *Rho* transcriptional repression (*Figure 2b*). We then analyzed the transcriptional perturbation in response to the AAV retinal gene transfer of ZF6-DB by determining the differentially expressed genes (DEGs). Remarkably, *in vivo* the ZF6-DB protein generated about ten-fold less transcriptional perturbation compared with the ZF6-KRAB protein (19 vs. 222 DEGs; *Figure 2e*). Notably, this magnitude of perturbation is twenty five-fold lower than that induced by the ablation of an endogenous rod-specific TF (NRL, 500 DEGs vs 19 DEGs, ZF6-DBD; [*Roger et al., 2014*]). Retinal-specific pathway analysis of DEGs showed that ZF6-DB–induced down-regulation is restricted to the *Rho* biochemical interactor *Gnat1* (*Palczewski, 2012*), and the up-regulation of 2 genes associated with acute phase inflammatory response, *alpha-2-macroglobulin (A2m)* and *glial fibrillary acidic protein (Gfap)* (*Figure 2c*). ZF6-KRAB induced the de-regulation of 17 retinal network associated genes (*Figure 2—figure supplement 1*). These results suggest that both ZF6-DB and the consequent *Rho* down-regulation marginally interferes with photoreceptor specific pathways, apart from *Gnat1* repression, and that the up-regulation of the inflammatory response genes may be due to the collapse of the retinal scaffold caused by Rho depletion. The intersection of retinal transcriptome changes between ZF6-DB and ZF6-KRAB showed that both drive similar perturbation in the expression of 16 genes, which represent 84% of the entire pool of ZF6-DB DEGs (*Figure 2e*). Consistently, both ZF6-DB and ZF6-KRAB generated similar functional effects, i.e. concordant up- or down- differential expression of these 16 shared genes (*Figure 2d*). These results suggest that both ZF6-DB and ZF6-KRAB bind to similar genomic targets. We next studied whether the differential transcriptional repression induced by ZF6-DB and ZF6-KRAB was due to similar biochemical binding properties for the ZF6-cis DNA target. Both ZF6-DB and ZF6-KRAB proteins bind the ZF6-cis *RHO* DNA target site with similar affinities (*Figure 2—figure supplement 2*). These data suggest that, despite the presence of an active canonical repressor domain, ZF6-KRAB generated *Rho* silencing by DNA binding. Indeed, ZF6-DB, being exclusively a DNA-binding protein identical to the DBD of ZF6-KRAB, showed similar *Rho* silencing effects but far less retinal transcriptional perturbations. This indicates that the specificity is conferred by both the engineered design of the DNA-binding on a genome-specific target (*Beerli and Barbas, 2002*) and the lack of the ED. In addition, this finding supports the notion that that ZF6-cis CRE is necessary for *Rho* expression genome-wide.

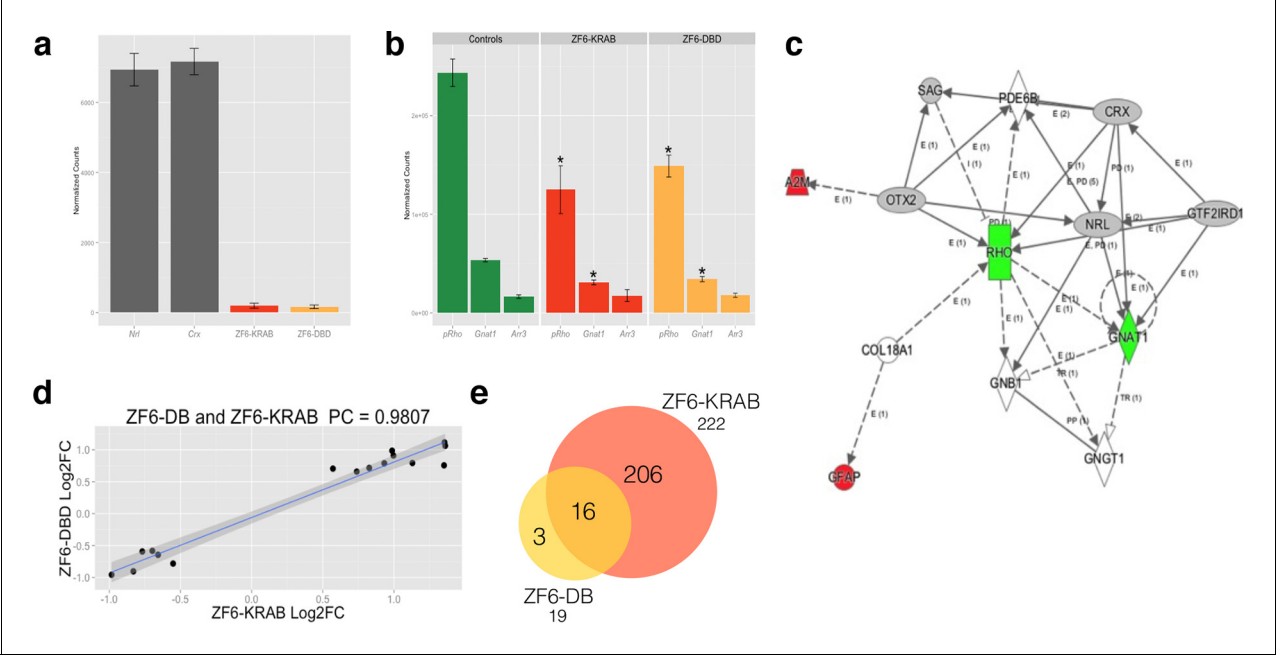

**Figure 2.** Photoreceptor delivery of ZF6-DB resulted in reduced genome-wide transcript perturbations. (a) RNA-Seq expression levels (Mean Normalized Counts) comparison between 2 endogenous TFs (*Crx* and *Nrl*) and the expression levels resulting from transduction of AAV8-CMV-ZF6-DB and AAV8-CMV-ZF6-KRAB, 15 days after retinal delivery (AAV8-CMV-ZF6-DB n= 6; AAV8-CMV-ZF6-KRAB n= 4 and 7 controls, non-transduced area). (b) *Rho* and rod *Gnat1* and Cone *Arrestin 3* expression levels in treated and control retina. (c) Ingenuity Pathway Analysis of DEGs after ZF6-DB AAV delivery in porcine retina showed a network of 13 genes. The 2 phototransduction genes *RHO* and *GNAT1* are shown in green (down-regulated) whereas the 2 genes associated with primary inflammatory response network, *A2M* and *GFAP*, are up-regulated (red). (d) Transcriptional activation and repression concordances among Log Fold Changes of the genes in common (*Swaroop et al., 2010*) between ZF6-DB and ZF6-KRAB (Pearson Correlation Test; PC=0.9787; p value << $1 \times 10^{-5}$). (e) Venn Diagrams, pairwise intersection of the 2 sets of Differentially Expressed Genes (DEGs). An adjusted p value (False Discovery Rate; FDR $\leq$ 0.1), without filtering on fold change levels, resulted in 19 and 222 DEGs, in ZF6-DBD and ZF6-KRAB treated retina, respectively. The intersection resulted significant by hypergeometric test (p value << $1 \times 10^{-5}$).

The following figure supplements are available for figure 2:

**Figure supplement 1.** Ingenuity Pathway Analysis on DEGs of ZF6-KRAB treated retina.

**Figure supplement 2.** Determination of the binding constants of ZF6-KRAB and ZF6-DB.

To test the functional activity of ZF6-DB in an adRP animal model, we used the *RHO-P347S* mouse (*Li et al., 1996*), which harbors the *P347S RHO* human mutant allele including the 20 DNA base pairs of the human genomic ZF6-cis sequence, whereas the murine *Rho* promoter sequence lacks the ZF6-Cis target (*Figure 1—figure supplement 1a*) (*Li et al., 1996*). Therefore, human-specific P347S *RHO* silencing by ZF6-DB (which does not affect murine *Rho* expression) may result in preservation of retinal function (*Mussolino et al., 2011a*). Strikingly, AAV8 vector delivery of ZF6-DB resulted in significantly higher functional protection in both the A- and B-wave components of electroretinography (ERG analysis) compared to ZF6-KRAB and AAV-GFP controls (*Figure 3a*). In addition, injection of either ZF6-KRAB or ZF6-DB in C57Bl/6 wild type retina did not produce detectable functional detrimental effects (*Figure 3b*). Thus, the higher ERG responses observed in ZF6-DB- compared to ZF6-KRAB-treated P347S mice should be further investigated.

To determine the therapeutic potential of DNA-binding-mediated silencing, we carried out the silencing-replacement strategy (*Kiang et al., 2005*) by coupling ZF6-DB with *RHO* replacement (human *RHO*, *hRHO* CDS) in order to complement *Rho* transcriptional repression in porcine retina (*Figure 4—figure supplement 1*). To achieve simultaneous photoreceptor transduction of both ZF6-DB and *hRHO*, we cloned the two expression cassettes into a single vector (DNA-binding repression and replacement, DBR-R, construct; *Figure 4a*). The key variables to achieve highly differential expression required for balanced simultaneous *RHO* repression and replacement are the vector

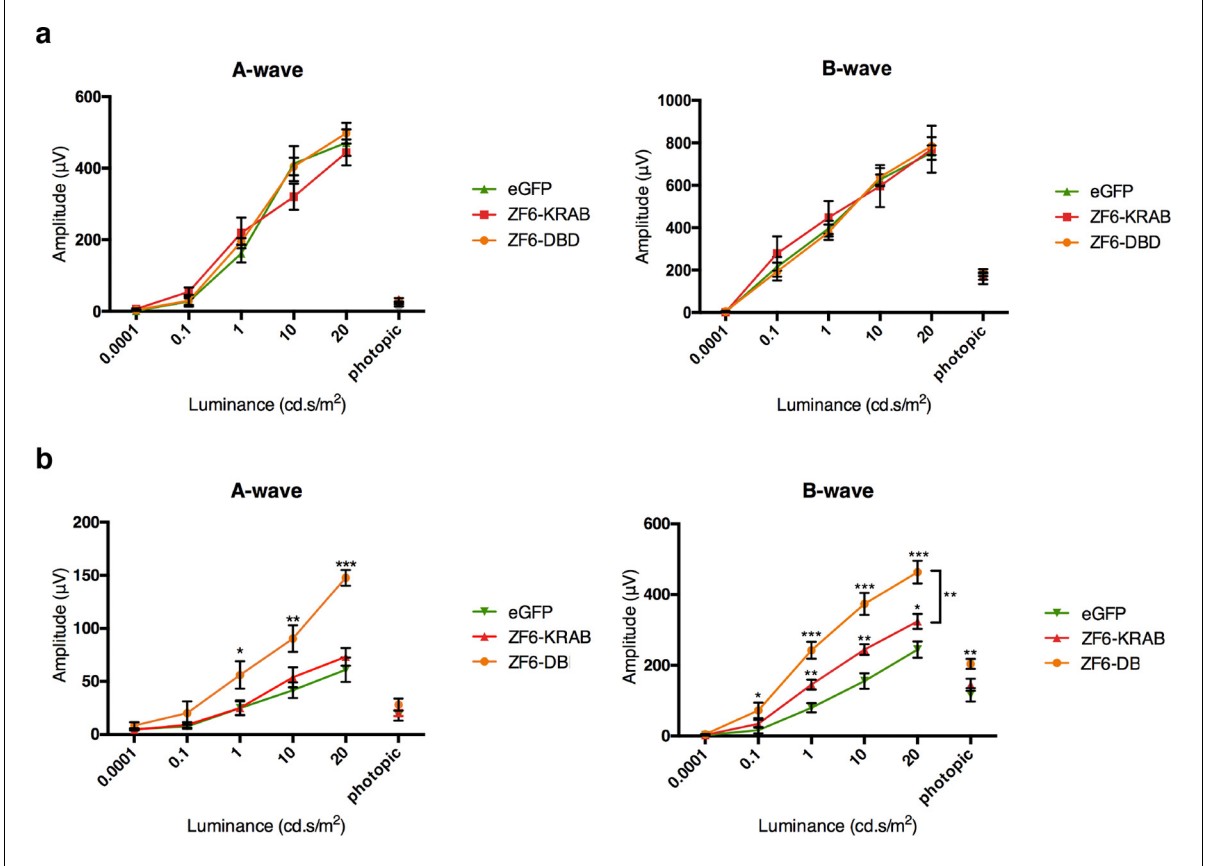

**Figure 3.** ZF6-DB DNA-binding protein preserves retinal function of the P347S adRP mouse model. (**a**) Electroretinography (ERG) analysis on P347S mice mice subretinally injected at post natal day 14 (PD14) with AAV8-CMV-ZF6-DB (n=10), AAV8-CMV-ZF6-KRAB (n=10), or AAV8-CMV-eGFP (n=10) and analysed at P30. Retinal responses in both scotopic (dim light) and photopic (bright light) showed that both A- and B-waves amplitudes, evoked by increasing light intensities, were preserved in both AAV8-CMV-ZF6-DB and ZF6-KRAB compared to eGFP control (**b**) A- and B-wave are shown for injected C57Bl/6 mice with ZF6-DB (n=4), ZF6-KRAB (n=4) and eGFP (n=4), independently. No functional impairment is observed for each construct. Error bars, means +/- s.e.m. *p<0.05, **p<0.01, ***p<0.001; two-tailed Student's t test.

dose and promoter strength. Indeed, ZF6-DB (~200 counts; RNA-seq expression levels) generated a decrease of about 100,000 *Rho* RNA-seq counts after transduction (~250,000 counts in controls vs. ~150,000 after treatment; *Figure 2a,b*). Thus, to ensure high and rod-specific *hRHO* replacement, we opted for a high vector dose and the strength of the *GNAT1* promoter (*Figure 1h*; [*Lee et al., 2010*]). To decrease ZF6-DB expression levels at high vector dose, while keeping rod-specificity, we both shortened the human *RHO* promoter and deleted the 5' sequence of the ZF6-DB target ZF6-Cis (*Figure 4—figure supplement 2*). We used $1 \times 10^{12}$ gc of vector of DBR-R (AAV8-RHOΔ-ZF6-DB-GNAT1-hRHO) to administer to porcine retina. As an internal control the contralateral eye received the previously used ZF6-DB vector (AAV8-CMV-ZF6-DB; at $1 \times 10^{10}$ gc; *Figure 1*). AAV8-CMV-eGFP was co-administrated to label the transduced area. Administration of the DBR-R vector resulted in rod-specific transcriptional repression of the porcine *Rho* (38%) and in concomitant replacement of the exogenous *hRHO* (68%), as assessed by transcripts, protein expression levels, and integrity of photoreceptor outer segments (*Figure 4b–d* and *Figure 4—figure supplement 3*). Notably, *Gnat1* transcript and protein (data not shown) levels demonstrated complementation, supporting a secondary down-regulation of *Gnat1* associated with *Rho* repression (*Figure 4b*).

In this study, we showed that photoreceptor genomic binding of a 20 bp-long DNA sequence by a synthetic DNA-binding protein dramatically reduces *Rho* expression. The combination of *Rho* transcriptional silencing and the restricted transcriptome perturbation induced by ZF6-DB, without the intrinsic repression activity contained in an ED, indicate that the local binding to ZF6-cis per se is the determinant of transcriptional repression, whereas the high specificity observed may result by both

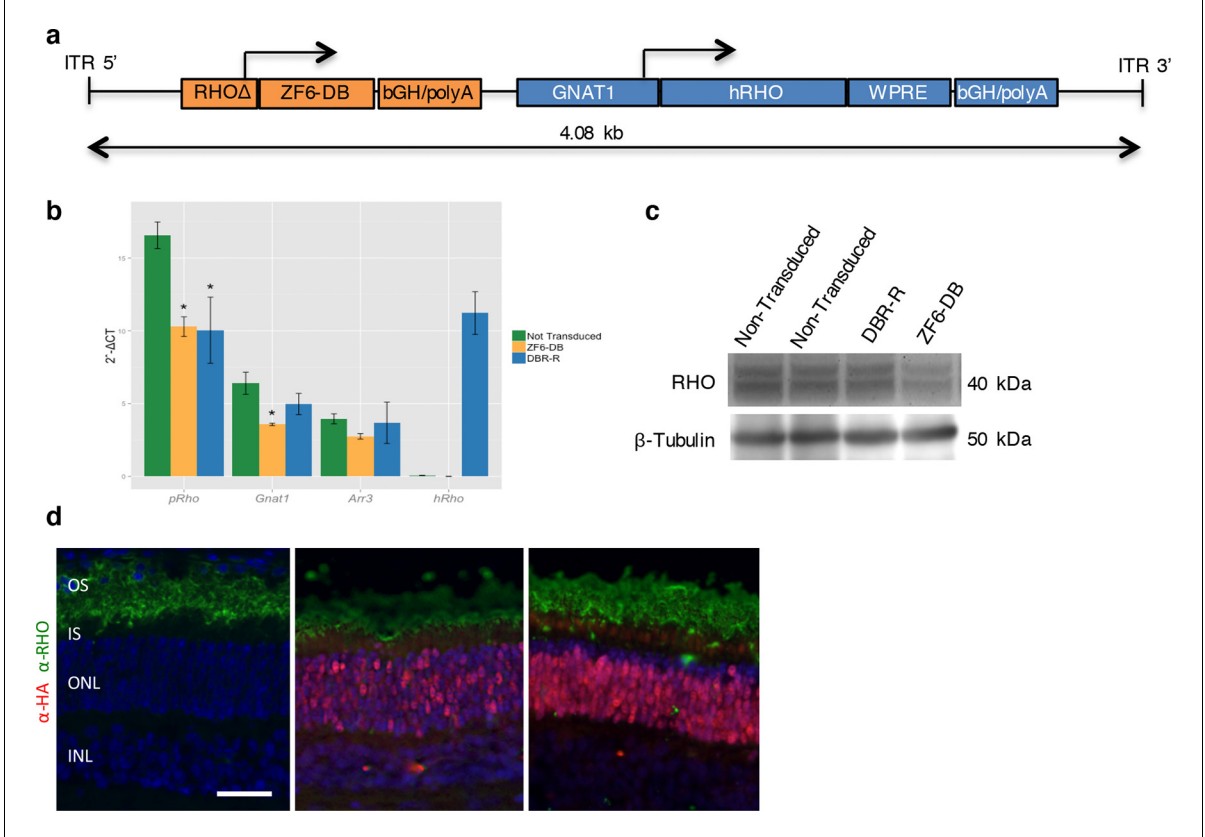

**Figure 4.** DNA-binding repression-replacement (DBR-R) of *Rho* in the porcine retina. (a) AAV8-RHOΔ-ZF6-DB-GNAT1-hRHO DBR-R construct features, including the two expression cassettes, RHOΔ-ZF6-DB encoding for both the DNA-binding repressor ZF6-DB (orange), and GNAT1-hRHO for human *RHO* for replacement (blue). The size (kb) of the construct is indicated as a bar. (b) qReal Time PCR, mRNA levels ($2^{-\Delta CT}$) 2 months after vector delivery of either AAV8-CMV-ZF6-DB (DBR; orange bars) or AAV8-RHOΔ-ZF6-DB-GNAT1-hRHO (DBR-R, blue bars) and non-transduced controls (green bars). *pRho*, porcine Rhodopsin; *Gnat1*, Guanine Nucleotide Binding Protein1; *Arr3*, Arrestin 3; *hRHO*, human *Rhodopsin*. The result is representative of two independent experiments. Error bars, means +/- s.e.m. ; *p<0.05, **p<0.01, ***p<0.001; two-tailed Student's t test. (c) Western blot analysis on the retina showed in b, c and d. (d) Immunofluorescence double staining with Rho (green) and HA-ZF6-DB (red) antibodies. Left panel, non-transduced control retina; middle panel, AAV8-CMV-ZF6-DB treated retina; left panel, AAV8-RHOΔ-ZF6-DB-GNAT1-hRHO DBR-R treated retina. OS, outer segment; IS, inner segment; ONL, outer nuclear layer; INL, inner nuclear layer; scale bar, 100 μm.

The following figure supplements are available for figure 4:

**Figure supplement 1.** Outline of the DNA-binding repressor-replacement (DBR-R) strategy.

**Figure supplement 2.** Strength and tissue specificity of RHOΔ promoter elements in murine retina.

**Figure supplement 3.** Cone morphological integrity after DNA-binding repression-replacement (DBR-R) subretinal delivery.

DNA-binding specificity (biochemical affinity) of ZF6-DB and the rod genomic context. The transcriptional repression mechanism of ZF6-DB binding likely relies on the interference occurring between TFs and local DNA sequence features within the *RHO* proximal promoter region (*Mitton et al., 2000*), which we showed here to be necessary to control *Rho* expression at the genomic level. The lack of known TFBSs and the low level of expression of ZF6-DB expressed in the photoreceptors, which was twenty-fold below the levels of photoreceptor specific TFs (*Crx* and *Nrl*), suggest that the molecular determinant of silencing may not be the simple displacement of key *RHO* TFs (*Mao et al., 2011*). We propose a model in which the molecular features of the DNA (loop, twisting, bending, for instance) may contribute to *Rhodopsin* transcriptional output. In this context, the DNA may be envisaged as not being exclusively the source of storage of functional information (protein coding and non-coding transcripts) or an inert DNA-binding protein harbor (i.e. positional information for TFs

DNA binding), but also as an intrinsically active operator of the transcriptional function. This contribution of DNA is supported by both the transcriptional repression upon the ZF6-cis deletion of 20 bp of DNA and intereference in trans by using a synthetic DNA binding protein that, not being encoded by the genome, may occupy a protein-free portion of the genome. It follows that in terms of signaling, the information source, the DNA, generates an output signal the RNA and eventually a protein (TF), whose final output functional activity is completed back by the DNA. Thus the information source, the DNA, becomes also integral part of the signaling (*Rhodopsin* transcriptional output). In this vision, to act (interfere) upstream the DNA, an external function is necessary, which is, in this study, the synthetic ZF6-DB carried by the vector.

From a therapeutic prospective, a relevant property of ZF6-DB DNA binding interference is the high rate of transcriptional silencing observed after in vivo gene transfer, which is consistent with canonical TFs mode of action (*Kiang et al., 2005*). DNA binding interference via ZF6-DB in transduced retina generated 45% *Rho* transcriptional repression, which reached 85% when rods were sorted, supporting its use for diseases requiring correction of a large number of affected cells, such as adRP and other Mendelian disorders due to gain-of-function mutations. Furthermore, *Rho* transcriptional silencing and its complete *RHO* replacement support in principle the use of the DBR-R constructs for treatment of any *RHO* mutation including those caused by a dominant negative mechanism (*Mao et al., 2011*). However, the 38% silencing and replacement observed may not yet be sufficient to achieve therapeutic efficacy/benefit in patients with adRP. Therefore, further development of this proof-of-concept will include optimization of the design of the silencing and replacement double construct (tuning the strength of the promoter elements), vector selection and dose, and the surgical approach. In conclusion, in vivo retinal gene transfer of an AAV vector (*Doria et al., 2013*; *Liang et al., 2001*; *Scatchard, 1949*) carrying a 22 kDa orthogonal (*Surace et al., 2005*) and gene-extrinsic noise-resistant (the permissive rod photoreceptor cell-specific environment; [*Li et al., 2011*]) synthetic protein acts as a transcriptional repressor, which results in a potent and specific silencing of the *Rho* gene upon binding to an essential *Rho* DNA element.

## Materials and methods

### Plasmid construction

The ZF6-DNA-binding domain (NΔ96 deletion mutant, ZF6-DB) was amplified by PCR from AAV2.1 CMV-ZF6-KRAB (*Mussolino et al., 2011a*) using primers ZF6-DBfw (TTGCGGCCGCATGATCGATC TGGAACCTGGCG) and ZF6-DBrv (AAGCTTTCAAGATGCATAGTCT). The PCR product was digested using NotI and HindIII restriction enzymes and cloned in pAAV2.1. The *hGNAT1* promoter was synthetized by Eurofins MWG based on *Lee et al. 2010* adding the 5'UTR. The fragment was cloned in pAAV2.1 using NheI and NotI restriction enzymes. The human Rhodopsin CDS was amplified by PCR from human retina cDNA using the hRHOfw (GCGGCCGCATGAATGGCACA-GAAGGCCC) e hRHOrv (AAGCTTTTAGGCCGGGGCCACCTG) primers and the PCR fragment was digested using NotI and HindIII restriction enzymes and cloned in pAAV2.1 plasmid under the control of *hGNAT1* promoter. The human rhodopsin short promoter (hRHO-short-(s), 164 bp from the transcription starting site (TSS) + 5'UTR), the human rhodopsin long promoter (hRHO-long, 796 bp from the TSS + 5'UTR), the human rhodopsin long promoter mutated of the ZF6-cis (hRHO-cis-del, 776 bp from the TSS lacking the bases -82 -62 from the TSS) and the human rhodopsin muted promoter (hRHO-s-ΔZF6, lacking the bases -84 -77 from the TSS) were generated by gene synthesis of Eurofins MWG and cloned in pAAV2.1 using NheI and NotI restriction enzymes. For the generation of DBR-R plasmid the Eurofins MWG synthetized the expression cassette RHOΔ-ZF6-DB-bGHpolyA (bovine growth hormone polyA) that we cloned in pAAV2.1 hGNAT1-hRHO using NheI restriction enzyme.

### AAV vector preparations

AAV vectors were produced by the TIGEM AAV Vector Core, by triple transfection of HEK293 cells followed by two rounds of $CsCl_2$ purification (*Auricchio et al., 2001*). For each viral preparation, physical titers [genome copies (GC)/mL] were determined by averaging the titer achieved by dot-blot analysis (*Doria et al., 2013*) and by PCR quantification using TaqMan (Applied Biosystems, Carlsbad, CA, USA).

## Vector administration and animal models

All procedures were performed in accordance with institutional guidelines for animal research and all of the animal studies were approved by the authors. P347S+/+ animals (*Mussolino et al., 2011a*; *Li et al., 1996*) were bred in the animal facility of the Biotechnology Centre of the Cardarelli Hospital (Naples, Italy) with C57Bl/6 mice (Charles Rivers Laboratories, Calco, Italy), to obtain the P347S+/- mice.

## Mice

Intraperitoneal injection of ketamine and medetomidine (100 mg/kg and 0.25 mg/kg respectively), then AAV vectors were delivered sub-retinally via a trans-scleral transchoroidal approach as described by Liang et al. (*Liang et al., 2001*).

## Pigs

Eleven-week-old Large White (LW) female piglets were utilized. Pigs were fasted overnight leaving water *ad libitum*. The anesthetic and surgical procedures for pigs were previously described (*Mussolino et al., 2011b*). AAV vectors were inoculated sub-retinally in the avascular nasal area of the posterior pole between the two main vascular arches, as performed in Mussolino et al. (*Mussolino et al., 2011b*). This retinal region is crossed by a streak-like region that extends from the nasal to the temporal edge parallel to the horizontal meridian, where cone density is high, reaching 20,000 to 35,000 cone cells mm$^2$. Each viral vector was injected in a total volume of 100 µl, resulting in the formation of a subretinal bleb with a typical 'dome-shaped' retinal detachment, with a size corresponding to 5 optical discs.

## Cloning and Purification of the proteins

DNA fragments encoding the sequence of the engineered transcription factors and ZF6-KRAB, to be expressed as maltose-binding protein (MBP) fusion were generated by PCR using the plasmids pAAV2.1 CMV-ZF6-KRAB and pAAV2.1 CMV-ZF6-DB as a DNA template. The following oligonucleotides were used as primers: primer 1, (GGAATTCCATATGGAATTCCCCATGGATGC) and primer 2, (CGGGATCCCTATCTAGAAGTCTTTTTACCGGTATG), for ZF6-KRAB primer 3, (GGAATTCCATATGCTGGAACCTGGCGAAAAACCG) and primer 4,(CGGGATCCCTATCTAGAAGTCTTTTTACCGGTATG) for ZF6-DB. All the PCR products were digested with the restriction enzymes NdeI and BamH1 and cloned into NdeI BamH1-digested pMal C5G (New England Biolabs, Ipswich, MA) bacterial expression vector. All the plasmids obtained were sequenced to confirm that there were no mutations in the coding sequences. The fusion proteins were expressed in the *Escherichia coli* BL21DE3 host strain. The transformed cells were grown in rich medium plus 0.2% glucose (according to protocol from New England Biolabs) at 37°C until the absorbance at 600 nm was 0.6–0.8, at which time the medium was supplemented with 200 µM ZnSO4, and protein expression was induced with 0.3 mM isopropyl 1-thio-β-D-galactopyranoside and was allowed to proceed for 2 hr. The cells were then harvested, resuspended in 1X PBS (pH 7.4), 1 mM phenylmethylsulfonyl fluoride, 1 µM leupeptin, 1 µM aprotinin, and 10 µg/ml lysozyme, sonicated, and centrifuged for 30 min at 27,500 relative centrifugal force. The supernatant was then loaded on amylose resin (New England Biolabs) according to the manufacturer's protocol. To remove the MBP from the proteins, bound fusion proteins as cleaved in situ on the amylose resin with Factor Xa (1 unit/20 µg of MBP fusion protein) in FXa buffer (20 mM Tris, pH 8.0, 100 mM NaC1, 2 mM CaC1$_2$) for 24–48 hr at 4°C and collected in the same buffer after centrifugation at 500 relative centrifugal force for 5 min. The supernatant containing the protein without the MBP tag was then recovered.

## Gel mobility shift analysis

The affinity binding costant of proteins for *hRHO* proximal promoter sequence was measured by a gel mobility shift assay by performing a titration of the proteins with the oligonucleotides. The purified proteins were incubated for 15 min on ice with hRHO 65 bp duplex oligonucleotide in the presence of 25 mM Hepes (pH 7.9), 50 mM KCl, 6.25 mM MgCl2, 1% Nonidet P-40, 5% glycerol. After incubation, the mixture was loaded on a 5% polyacrylamide gel (29:1 acrylamide/bisacrylamide ratio) and run in 0.5 TBE at 4°C (200 V for 4 hr). Protein concentration was determined by a modified version of the Bradford procedure. After electrophoresis, the gel was stained with the fluorescent dyes

SYBR Green I Nucleic acid gel stain (Invitrogen, Carlsbad, CA) to visualize DNA. 2.5 µM of the ZF6-KRAB protein was incubated with increasing concentrations (130, 135, 145, 150, 165, 170, 175, 180, 190, and 200 nM, respectively) of the duplex hRHO 65 bp, an apparent higher protein concentration (2.5 µM) was required likely because not all the protein sample was correctly folded. In the case of ZF6-DB, 1.5 µM of the protein was incubated with increasing concentrations (145, 150, 170, 175, 195, 210, 220, 225, 240, and 250 nM, respectively) of the duplex hRho 65 bp. Scatchard analysis of the gel shift binding data was performed to obtain the Kd values (25). All numerical values were obtained by computer quantification of the image using a Typhoon FLA 9500 biomolecular imager (GE Healthcare Life Sciences).

## qReal time PCR

RNAs from tissues were isolated using RNAeasy Mini Kit (Qiagen, Germany), according to the manufacturer protocol. cDNA was amplified from 1 µg isolated RNA using QuantiTect Reverse Transcription Kit (Qiagen), as indicated in the manufacturer instructions.

The PCRs with cDNA were carried out in a total volume of 20 µl, using 10 µl LightCycler 480 SYBR Green I Master Mix (Roche, Switzerland) and 400 nM primers under the following conditions: pre-Incubation, 50°C for 5 min, cycling: 45 cycles of 95°C for 10 s, 60°C for 20 s and 72°C for 20 s. Each sample was analysed in duplicate in two-independent experiments. Transcript levels of pig retinae were measured by quantitative Real Time PCR using the LightCycler 480 (Roche) and the following primers: pRho_forward (ATCAACTTCCTCACGCTCTAC) and pRho_reverse (ATGAAGAGG TCAGCCACTGCC), pGnat1_forward (TGTGGAAGGACTCGGGTATC) and pGnat1_reverse (GTC TTGACACGTGAGCGTA), pArr3_forward (TGACAACTGCGAGAAACAGG) and pArr3_reverse (CACAGGACACCATCAGGTTG). humanRho_forward (TCATGGTCCTAGGTGGCTTC), humanRho_reverse (ggaagttgctcatgggctta) and eGFP_forward (ACGTAAACGGCCACAAGTTC) and eGFP_reverse (AAGTCGTGCTGCTTCATGTG). All of the reactions were standardized against porcine Actβ using the following primers: Act_Forward (ACGGCATCGTCACCAACTG) and Act_reverse (CTGGG TCATCTTCTCACGG).

## Immunostaining

Frozen retinal sections were washed once with PBS and then fixed for 10 min in 4% PFA. Sections were immerse in a retrieval solution (0,01 M citrate buffer, pH 6.0) and boiled three times in a microwave. After the blocking solution (10% FBS, 10% NGS, 1% BSA) was added for 1 hr. The primary antibody mouse anti-HA (1:300, Covance) was diluted in a blocking solution and incubated overnight at 4°C. The secondary antibody (Alexa Fluor® 594, anti-mouse 1:1000, Molecular Probes, Invitrogen, Carlsbad, CA) has been incubated for 1 hr. Vectashield (Vector Lab Inc., Peterborough, UK) was used to visualize nuclei. Frozen retinal sections were permeabilized with 0.2% Triton X-100 and 1% NGS for 1 hr, rinsed in PBS, blocked in 10% normal goat serum (NGS), and then incubated overnight at 4°C with rabbit human cone arrestin (hCAR) antibody, kindly provided by Dr. Cheryl M. Craft (Doheny Eye Institute, Los Angeles, CA) diluted 1:10000 in 10% NGS. After three rinses with 0.1 M PBS, sections were incubated in goat anti-rabbit IgG conjugated with Texas red (Alexa Fluor 594, anti-rabbit 1:1000, Molecular Probes, Invitrogen, Carlsbad, CA) for 1 hr followed by three rinses with PBS. Frozen retinal sections were permeabilized with 0.1% Triton X-100, rinsed in PBS, blocked in 20% normal goat serum (NGS), and then incubated overnight at 4°C in a mouse anti-1D4 rhodopsin antibody diluted 1:500 in 10% NGS. After three rinses with 0.1 M PBS, sections were incubated in goat anti-mouse IgG conjugated with Texas red (Alexa Fluor® 594, anti-mouse 1:1000, Molecular Probes, Invitrogen, Carlsbad, CA) for 1 hr followed by other three rinses with PBS. Sections were photographed using either a Zeiss 700 Confocal Microscope (Carl Zeiss, Oberkochen, Germany) or a Leica Fluorescence Microscope System (Leica Microsystems GmbH, Wetzlar, Germany). *Triple-immunostaining for anti-HA, anti-GNAT1, and anti-Rhodopsin antibody.* Frozen retinal sections were washed once with PBS and then fixed for 10 min in 4% PFA. Sections were immerse in a retrieval solution (0,01 M sodium citrate buffer, pH 6.0) and boiled three times in a microwave. After the blocking solution (10% FBS, 10% NGS, 1% BSA) was added for 1 hr. The two primary antibody mouse anti-HA (1:300, Covance) and rabbit GαT1 (Santacruz Biotechnology), were diluted in a blocking solution and incubated overnight at 4°C. The secondary antibodies (Alexa Fluor 594, anti-mouse 1:800, Molecular Probes, and Alexa Fluor 488, anti-rabbit 1:500, Molecular Probes, Invitrogen,

Carlsbad, CA) have been incubated for 1 hr, followed by three rinses with PBS. After the slides were incubated in blocking solution (10% NGS) for 1 hr and then incubated O.N. with primary antibody mouse- 1D4 (1:500, Abcam). The secondary antibodies (Alexa Fluor 405, anti-mouse 1:200, Molecular Probes, Invitrogen). Sections were photographed using a Leica Fluorescence Microscope System (Leica Microsystems GmbH, Wetzlar, Germany).

## Western blot analyses

Western blot analysis was performed on retinae, which were harvested. Samples were lysed in hypotonic buffer (10 mM Tris-HCl [pH 7.5], 10 mM NaCl, 1,5 mM $MgCl_2$, 1% CHAPS, 1 mM PMSF, and protease inhibitors) and 20 µg of these lysates were separated by 12% SDS-PAGE. After the blots were obtained, specific proteins were labeled with anti-1D4 antibody anti-Rhodopsin-1D4 (1:1000; Abcam, Cambridge, MA), and anti-β-tubulin (1:10000; Sigma-Aldrich, Milan, Italy) antibodies.

## Chromatin immunoprecipitation experiments (ChIP)

For ChIP experiments, HEK293 cells were transfected by $CaCl_2$ with pAAV2.1 CMV-ZF6-KRAB, pAAV2.1 CMV-ZF6-DB or pAAV2.1 CMV-eGFP. The cells are harvested after 48 hr. ChIP was performed as follow: cells were homogenized mechanically and cross linked using 1% formaldehyde in PBS at room temperature for 10 min, then quenched by adding glycine at final concentration 125 mM and incubated at room temperature for 5 min. Cells were washed three times in cold PBS 1X then cells were lysed in cell lysis buffer (Pipes 5 mM pH 8.0, Igepal 0.5%, KCl 85 mM) for 15 min. Nuclei were lysed in nuclei lysis buffer (Tris HCl pH8.0 50 mM, EDTA 10 mM, SDS 0.8% ) for 30 min. Chromatin was shared using Covaris s220. The shared chromatin was immunoprecipitated over night with anti HA ChIP grade (Abcam, ab 9110, Cambridge, MA). The immunoprecipitated chromatin was incubated 3 hr with magnetic protein A/G beads (Invitrogen, Carlsbad, CA). Beads were than washed with wash buffers and DNA eluted in elution buffer (Tris HCl pH 8 50 mM, EDTA 1 mM, SDS 1%). Then Real Time PCR was performed using primers on rhodopsin TSS, hRHOTSSFw (TGACC TCAGGCTTCCTCCTA) and hRHOTSSRv (ATCAGCATCTGGGAGATTGG), trasducin 1 TSS, hGNAT1TSSFw (CAGCCCTGACCCTACTGAAC) and hGNAT1TSSRv (CAACCGCTGACTCTGCACT), arrestin 3 TSS, hArr3TssFw (CCTGCTGTGCACATAAGCTG) and hArr3TssRv (CGTGTCCCACTCCAA TCTCT), and β-tubulin TSS, hTUBTSSFw (TCCTGTACCCCCAAGAACTG) and hTUBTSSRv (GC TGCAAAATGAAGTGACGA).

## FACS rods sorting

Co-injected porcine retina with AAV8-CMV-ZF6-DB (dose $5x10^{10}$ gc)and AAV8-GNAT1-eGFP (dose $1x10^{12}$ gc) were disaggregated using Papain Dissociation System (Worthington biochemical corporation) following the manufacturers protocol. Dissociated retinal cells were analysed using BD FACSAria at IGB (Institute of Genetic and Biophysic "A. Buzzati-Traverso") FACS Facility and sorted, dividing eGFP positive cells (rods) from eGFP negative fraction.

## Electrophysiological testing

The method is as described (*Surace et al., 2005*). Brifley, mice were dark reared for 3 hr and anesthetized. Flash electroretinograms (ERGs) were evoked by 10-ms light flashes generated through a Ganzfeld stimulator (CSO, Costruzione Strumenti Oftalmici, Florence, Italy) and registered as previously described. ERGs and b-wave thresholds were assessed using the following protocol. Eyes were stimulated with light flashes increasing from $-5.2$ to $+1.3$ log cd*s/$m^2$ (which correspond to $1 \times 10^{-5.2}$ to 20.0 cd*s/$m^2$) in scotopic conditions. The log unit interval between stimuli was 0.3 log from $-5.4$ to 0.0 log cd*s/$m^2$, and 0.6 log from 0.0 to $+1.3$ log cd*s/$m^2$. For ERG analysis in scotopic conditions the responses evoked by 11 stimuli (from $-4$ to $+1.3$ log cd*s/$m^2$) with an interval of 0.6 log unit were considered. To minimize the noise, three ERG responses were averaged at each 0.6 log unit stimulus from $-4$ to 0.0 log cd*s/$m^2$ while one ERG response was considered for higher (0.0$-+1.3$ log cd*s/$m^2$) stimuli. The time interval between stimuli was 10 s from $-5.4$ to 0.7 log cd*s/$m^2$, 30 s from 0.7 to $+1$ log cd*s/$m^2$, or 120 s from $+1$ to $+1.3$ log cd*s/$m^2$. a- and b-waves amplitudes recorded in scotopic conditions were plotted as a function of increasing light intensity (from $-4$ to $+1.3$ log cd*s/$m^2$, *Figure 3*). The photopic ERG was recorded after the scotopic session

by stimulating the eye with ten 10 ms flashes of 20.0 cd*s/m$^2$ over a constant background illumination of 50 cd/m$^2$.

## RNASeq library preparation, sequencing and alignment

The 17 libraries were prepared using the TruSeq RNA v2 Kit (Illumina, San Diego, CA) according to manufacturer's protocol. Libraries were sequenced on the Illumina HiSeq 1000 platform and in 100-nt paired-end format to obtain approximately 30 million read pairs per sample. Sequence Reads were trimmed using Trim Galore! software (v.0.3.3), that trims low-quality ends and removes adapter from reads, using a Default Phred score of 20. To obtain a precise estimation of this yet uncharacterized tissue, the 17 libraries were aligned against the full transcriptome for *Sus scrofa* (Pig) as provided by ENSEMBL (*SusScrofa* 10.2.73). The GTF included the sequences for the 20 canonical chromosomes plus 4563 scaffolds, and counted 30.567 transcripts plus the sequences for the 3 exogenes used in the analysis (the 2 TRs and eGFP). Alignment was performed with RSEM (v.1.2.11) (*Li et al., 2011*) with default parameters. The resulting expected counts (the sum of the posterior probability of each read coming from a specific transcript over all reads) were used for subsequent analysis.

## Differential expression analysis

The dataset was composed of 17 samples and 25.325 genes, divided in 3 experimental groups: 7 Controls, 4 ZF6-KRAB-treated, 6 ZF6-DB-treated.

We analyzed the data following the standard Differential Expression Analysis Pipeline with DESeq2 R/Bioconductor package (v.1.8.1) (*Love et al., 2014*), filtering and normalizing all libraries together. We filtered low tag counts retaining those which had 1 CPM in at least 3 samples.

We fitted a unique Generalized Linear Model (GLM) with 1 factor and 3 levels (Control, ZF6-KRAB-treated, ZF6-DB-treated). Differentially expressed genes were obtained out of the 2 contrasts (each treatment compared to the controls), an adjusted pvalue (FDR) of less than or equal to 0.1 was considered significant. We observed the expected upregulation of the exogenous genes used for the treatment (ZF6-KRAB, ZF6-DB, eGFP) and for further evaluations we didn't take into account their differential expression.

## Functional concordance

The 16 genes in common between the ZF6-DB and ZF6-KRAB DEGs were tested for functional concordance using Pearson product moment correlation coefficient (*cor.test*, R package *stats* v.3.2.0) (*Huber et al., 2015*; *R Core Team, 2015*).

Two named numeric vectors, one for each condition, containing the Log fold changes values of the 16 genes were tested for association with *cor.test* default function, method = 'Pearson'.

## Heatmap

A manually curated list of Human Gene IDs including representative Retinal Markes and a subset of Retina Disease Genes (*Daiger BR et al., 1998*) was used to show the interference power of the 2 TRs with the overall regulatory circuitry. The Human IDs were used to retrieve their homolog Porcine genes, if present. Genes with duplicated homolog in the *Sus scrofa* genome were included in the list (genes tagged with_1 in *Figure 2*).

## Data management

All the analyses, except for the reads quality filtering, alignment and expression estimates, were performed in the R statistical environment (v.3.2.0) (*Huber et al., 2015*; *R Core Team, 2015*). Plots were generated with ggplot2 R/Bioconductor package (v.1.0.1) (*Wickham, 2009*).

## Statistical analyses

Data are presented as mean ± Error bars indicate standard error mean (SEM). Statistical significance was computed using the Student's two-sided *t*-test and *p*-values <0.05 were considered significant. No statistical methods were used to estimate the sample size and no animals were excluded.

## Acknowledgements

We thank TIGEM NGS facility, TIGEM Bioinformatic core; TIGEM vector core for vector production; we thank Maria Matarazzo, Maurizio D'Esposito and Floriana Della Ragione for scientific support for ChIP experiments. We thank Diego di Bernardo, Graciana Diez-Roux, Antonella De Matteis, Alberto Auricchio, Cathal Wilson, Sandro Banfi, Nicola Brunetti-Perri, Robert Blelloch and Alison Forrester for discussions. We thank Stefano Carotenuto for the cartoon drowning. This work was supported by the European Research Council/ERC Grant Agreement No. 311682 'Allelechoker' and the Italian Telethon Foundation Grant TMESMT211TT.

## Additional information

### Funding

| Funder | Grant reference number | Author |
| --- | --- | --- |
| European Research Council | 311682 | Enrico Maria Surace |
| Fondazione Telethon | TMESMT211TT | Enrico Maria Surace |

The funders had no role in study design, data collection and interpretation, or the decision to submit the work for publication.

### Author contributions

SB, EM, NdP, FC, Acquisition of data, Analysis and interpretation of data, Drafting or revising the article; MR, Acquisition of data, Analysis and interpretation of data; MS, ML, AC, Acquisition of data; MLB, Coordinated the in vivo studies in large animals; CG, Assisted the subretinal injections in large animals; SR, Performed subretinal injections in large animals; FS, Supervised the subretinal injections and provided reagents for large animal studies; EMS, Conception and design, Drafting or revising the article

### Author ORCIDs

Enrico Maria Surace, http://orcid.org/0000-0002-2975-942X

### Ethics

Animal experimentation: All procedures were performed in accordance with institutional guidelines for animal research and all of the animal studies were approved by the authors. The protocol was approved by Italian Ministry for Health (IACUC protocols #114/2015-PR).

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
