## [Decision Letter]

Thank you for submitting your work entitled "Gene-targeted transcriptional silencing by DNA-binding" for consideration by *eLife*. Your article has been reviewed by three peer reviewers, and the evaluation has been overseen by a Reviewing Editor (Jeremy Nathans) and a Senior Editor.

The following individual involved in review of your submission has agreed to reveal their identity: Eric Pierce (peer reviewer).

The reviewers have discussed the reviews with one another and the Reviewing Editor has drafted this decision to help you prepare a revised submission.

The manuscript has been reviewed by three expert reviewers, and their assessments together with my own (Reviewing editor), form the basis of this letter.

All of the reviewers were impressed with the importance and novelty of your work, and we would like to see a revision that addresses the issues raised.

I am including the three reviews (lightly edited) at the end of this letter, as there are some specific and useful suggestions in them that will not be repeated in the summary here. Also, we think that this manuscript would be a better fit as a "Tools and Resources" paper rather than a regular research paper.

The main issues, which I have extracted from the reviews and the subsequent discussion between the reviewers and me, are:

1) Revisions to the text that focus the story on use of the ZF6-DB to suppress RHO expression. This could be accomplished by changing the title, and revising the Introduction and Discussion. We would favor moderating the claims about potential clinical use of this approach since the repression-replacement approach wasn't tested in a disease model.

2) There is a question of ZF6-DB specificity and efficiency. Please provide the porcine, human, and mouse RHO TSS sequences for comparison, since it appears that the ZF6-DB recognizing hRHO TSS was used in these models regardless of any differences between the sequences. Does ZF6-DB recognize the porcine and mouse sequence? As to clinical therapeutic potential, how do the pig and human repression efficiencies compare? Is the 35% repression of Rho expression sufficient for dominant disease? These are important questions that could be addressed by supplying the sequences and the related data.

3) It is unclear why cone arrestin increases in Arr3. This suggests that there may be off-target effects. Does ZF6-DB also bind to the Arr3 TSS region? ChIP data should be supplied to address off-target issues.

4) Figure 3 would benefit from the presentation of wild type mouse data. For example, eGFP, ZF6-KRAB, and ZF6-DB treatments of wild type mice to serve as a point of comparison and investigate the efficiency of ZF6-DB suppression of human and mouse rhodopsin respectively.

5) Regarding the repression-replacement experiment described in Figure 4, it is not clear why a different promoter (CMV) was used in the control ZF6-DB vector than that used for the ZF6-DB cassette (modified RHO promoter) in the in the experimental ZF6-DB – RHO expression vector.

6) The level of repression of the RHO gene achieved in the repression-replacement experiment described in Figure 4 is moderate. It is likely that greater than 35% suppression of a mutant RHO allele would be needed to have a therapeutic effect for patients affected by retinal degeneration due to gain-of-function mutations in RHO. While the data reported in Figure 1 suggest that repression of RHO expression in transduced photoreceptor cells is closer to 90%, it is unclear if suppressing RHO expression in 35% of photoreceptor cells would be therapeutic. Additional experiments to optimize the dose response and evaluate the potential of this approach for therapeutic use are needed to support the claim that a ZF6-DB based approach has the potential to be used clinically.

7) Clarity of presentation as indicated below by reviewers #2 and #3.

We look forward to receiving the revised version of your manuscript.

*Reviewer #1:*

Summary: Botta et al. report a method for gene-specific transcriptional silencing using an engineered DNA-binding domain, "ZF6-DB." AAV-mediated expression of ZF6-DB in photoreceptors resulted in blockade of Rho expression, leading to therapeutic effects in animal models with gain-of-function cis- Rho mutations, such as autosomal dominant retinitis pigmentosa.

Feedback:

Experiment 1-showed the 20 bp in ZF6-cis is required for Rho expression via eGFP reporter assay. ChIP showed ZF6-DB binds well to CRE. Put ZF6-DB into porcine retina using AAV and measured Rho transcripts, which were decreased. Finally, they compared ZF6-DB with eGFP expression levels

Criticism: n = 2 is a low number. They should supplement their claims with additional data points.

Experiment 2-Compared Rho transcription levels between ZF6-KRAB and ZF6-DB, found greater repression using ZF6-DB and less transcriptional perturbation.

Criticism: They claim that ZF6-KRAB and ZF6-DB bind to same genomic targets, but they didn't explain or otherwise address the possibility that the difference in the transcription perturbations may in fact be caused by off-targeting effects or the down-regulation of RHO expression. Should perform ChIP to confirm the two are binding to the same region (as well as Crx and Nrl). Figure 2 is confusing: comparing before and after ZF6-DB treatments, rather than different constructs would be more helpful.

Experiment 3-Compared the ERG a and b waves in the P347S RP murine model and saw much greater rescue in ZF6-DB compared with ZF6-KRAB or eGFP.

Criticism: they should further explore the reason that ZF6-DB has such a greater rescue than ZF6-KRAB despite their similar binding affinity (according to Figure 2—figure supplement 2). This is a surprising discrepancy that merits exploration and explanation. It would also be helpful for them to repeat this experiment in wild type animals to see if ZF6-DB also represses RHO expression and compare the percentage of repression in wild type and R347S alleles.

Experiment 4-Created construct with ZF6-DB and hRHO and found suppression of mutant mouse Rho and expression of the hRHO.

Criticism: unclear why they didn't consistently use the CMV promoter and opted for the truncated Rho promoter, despite similar repression levels of RHO. Any rescue beyond P30?

*Reviewer #2:*

This manuscript describes the use of a synthetic DNA binding protein to interfere with expression of the RHO gene. The experiments described build on prior work in which a hybrid protein consisting of a DNA binding domain coupled to a KRAB repressor domain was used to repress expression of the RHO gene. In this study, the activity of the DNA binding domain alone was evaluated.

It seems that the manuscript attempts to achieve two goals, and does not completely succeed for either. One focus is the potentially general idea that DNA binding proteins alone targeted to regulatory elements can be used to regulate expression of target genes. The data presented regarding the ZF6-DB for RHO support this idea, but without testing this hypothesis for other genes, the potential to use this approach more generally remains in question. The second focus is the idea that repression of RHO could be used therapeutically in patients with retinal degeneration caused by dominant, gain-of-function mutations in this gene. Since the ZF6-DB mediated repression of RHO expression isn't specific for the mutant allele, a repression-replacement strategy would be needed in patients. The tests of this approach described in the manuscript show some efficacy, but fall short of being convincing.

Specific comments are as follows:

1) Additional studies of the ZF6-DB approach with 1 or more additional genes would be needed to support the hypothesis that DNA binding proteins can be used broadly to regulate gene expression. Such data needs to be added to the manuscript, or the manuscript needs to be revised to focus on the specific application of this approach to the RHO gene.

2) The level of repression of the RHO gene achieved in the repression-replacement experiment described in Figure 4 is moderate. It is likely that greater than 35% suppression of a mutant RHO allele would be needed to have a therapeutic effect for patients affected by retinal degeneration due to gain-of-function mutations in RHO. While the data reported in Figure 1 suggest that repression of RHO expression in transduced photoreceptor cells is closer to 90%, it is unclear if suppressing RHO expression in 35% of photoreceptor cells would be therapeutic. Additional experiments to optimize the dose response and evaluate the potential of this approach for therapeutic use are needed to support the claim that a ZF6-DB based approach has the potential to be used clinically.

3) Along these lines, the experiment testing the activity of the ZF6-DB in the transgenic RHO-P347S mice is helpful, but does not support the use of the ZF6-DB mediated RHO repression approach in patients, since the mutant human transgene is present on a mouse background. It would be better to test the activity of the ZF6-DB in a model with a mutant mouse allele, and include the complete repression-replacement approach in these studies.

4) The ChIP experiment is helpful to show occupancy of the RHO CRE by ZF6-DB. It would be helpful to report what other sequences were detected in the ChIP experiment. That is, what else does the RHO ZF6-DB bind to? Additional ChIP data to address the question of what proteins normally occupy the RHO regulatory element would also be helpful.

5) The repression of GNAT1 expression observed following treatment with the RHO ZF6-DB is interesting. Is there a similar CRE upstream of the GNAT1 gene? Or is some other mechanism responsible to the linked regulation of expression, as suggested by the data in Figure 4? Further exploration of this issue is warranted.

6) The relative specificity of the ZF6-DB activity compared to that observed using the hybrid ZF6-DB-KRAB ED protein is impressive. Other studies have used more than 30 million sequence reads to sample gene expression by the neural retina thoroughly. It would be helpful to know why 30 million reads were used in this study, and if this provides complete sampling of retinal transcripts.

7) The discussion about the mechanism of ZFN-DB activity is interesting, but is beyond the data available in the current manuscript. The Discussion would be better focusing on interpreting the data in the manuscript.

*Reviewer #3:*

Botta and colleagues report on in vivo targeted silencing of rhodopsin expression by transcription factor proteins -with and without effector domain (ED)- engineered to recognize a 20 bp-cis-regulatory element of the proximal region of the rhodopsin promoter. They provide data which support the view that both engineered transcription factor proteins, with and without ED, suppresses rhodospin expression by DNA binding with similar affinities to the 20 bp-cis-regulatory element. They show that the engineered transcription factor lacking the effector domain silences rather specifically the expression of rhodopsin gene as determined by transcriptome analyses. They report that AAV-based delivery of the engineered rhodopsin-specific transcription factor protein without ED to the retina of a transgenic mouse with a human rhodopsin gene harboring a dominant negative RHO mutation prevented retinal degeneration. Finally, as a first step towards the use of this original strategy in the clinics, they nicely demonstrate that AAV-based delivery of pig-specific rhodospin silencing transcription factor protein without ED and of human Rhodopsin permitted preservation of retinal cells which deprived of pig-rhodopsin expressed the human counterpart.

This is very nicely designed and executed study with real therapeutic promises. I have no major concern regarding the scientific aspects of this sound work.

I have however a major critic regarding the writing of the manuscript which makes this beautiful piece of work sometimes very hard to follow. Here are some points to clarify:

1) Figure 1 would suggest to show clearly which nucleotide corresponds to +1 and which is the -84 -65 ZF6-cis sequence and to provide a legend mentioning each of the CRE that appear on the scheme.

2) Figure 1 are a little confusing as in 1C the green corresponds to the e-GFP whereas in 1E that is just below the green labels the rhodopsin.

3) Anti-HA is used both in immunochemistry and CHIP analysis but it is not clearly mentioned that the constructs which were used encode this flag.

4) CHIP-seq analysis: I would suggest to precise before giving the results a short sentence describing what was done, i.e. AAV-based delivery of constructs encoding eGFP or silencing transcription factor protein with or without ED in HEK293 cells, immunoprecipitation using the HA tag and RT-PCR amplification using primers flanking (where are they by the way?). The supporting figure is obscure. What are the MOCKs? Why is there no mention of the results using the CMV-ZF6-KRAB construct?

In these experiments AAV2.1vectors were used whereas some others used AAV8. Not all the readers are aware of their tropism differences.

5) Transcriptomic analyses. Despite the importance of this analysis I have been unable to find in the manuscript reference to what was exactly done. How were the transduced cells collected? Were the transduced regions microdissected and then the rods FACs sorted? How were the mRNA outputs from the transduced versus the untransduced regions matched?

Overall, I would suggest the authors to submit their revised manuscript to a colleague that is not involved in this work to ensure clarity.

[Editors' note: further revisions were requested prior to acceptance, as described below.]

Thank you for resubmitting your work entitled "RHODOPSIN-targeted transcriptional silencing by DNA-binding" for further consideration at *eLife*. Your revised article has been favorably evaluated by a Senior editor and Jeremy Nathans as the Reviewing editor. The manuscript has been improved but there are some remaining issues that need to be addressed before acceptance, as outlined below:

It looks substantially improved, but there is one aspect of the writing that needs work: various words or phrases that imply an absolute or extreme conclusion are over-stating the data. For example, in the Abstract is the phrase "complete gene silencing". The data do not support the use of the word "complete" – indeed, quantitative biological data only rarely support the use of that word. The best one can say of any observation of this type is that the signal is below the limit of detection. I suggest substituting the word "efficient" for "complete". Similarly, in the title of the legend to Figure 1 is the phrase "abolishes Rho expression". I suggest changing this to "dramatically reduces Rho expression". Please go through the manuscript carefully to make changes of this sort. My view is that your data speaks for itself, and that the manuscript should be carefully written so as to avoid over-stating the data.

---

## [Author Response]

*The main issues, which I have extracted from the reviews and the subsequent discussion between the reviewers and me, are: 1) Revisions to the text that focus the story on use of the ZF6-DB to suppress RHO expression. This could be accomplished by changing the title, and revising the Introduction and Discussion. We would favor moderating the claims about potential clinical use of this approach since the repression-replacement approach wasn't tested in a disease model.*

We changed the title, revisited the Abstract and the Discussion to focus on RHO transcriptional repression by DNA binding. In addition, we now moderated the claim about the potential clinical use in both the Abstract and the Discussion.

*2) There is a question of ZF6-DB specificity and efficiency. Please provide the porcine, human, and mouse RHO TSS sequences for comparison, since it appears that the ZF6-DB recognizing hRHO TSS was used in these models regardless of any differences between the sequences. Does ZF6-DB recognize the porcine and mouse sequence? As to clinical therapeutic potential, how do the pig and human repression efficiencies compare? Is the 35% repression of Rho expression sufficient for dominant disease? These are important questions that could be addressed by supplying the sequences and the related data.*

We provided the porcine, human, and mouse RHO TSS sequences for comparison (Figure 1—figure supplement 1). As shown now in the figure (Figure 1—figure supplement 1) human and porcine RHO TSS share 19 out of 20 DNA bp whereas, the mouse RHO TSS sequence diverges from that of human. The differences in sequence present in the mouse RHO TSS make it resistant to the activity of the ZF6-DB. As now shown (Figure 3), the use of the ZF6-DB in wild-type mice did not result in any detectable effects as assessed by ERG analysis, thus supporting the specificity of the effect of ZF6-DB on both the human RHO promoter (present as a transgene in the P347S mouse model of adRP) and on the porcine RHO promoter.

Regarding the 35% repression of Rho expression in the whole porcine retina we agree that it might not be sufficient to treat a dominant disease such as adRP, although in singularly transduced cell (sorted rods) the repression almost reached 90%. We stated in the Discussion that the study provided a proof of concept and there is room of improvement the efficiency of the strategy on the whole retina scale in order to ensure therapeutic benefit.

We added one additional retinal sample in the silencing and replacement experiment which now has n= 3 (Figure 4).

We added one additional rod sorted sample in the silencing experiment which now has n= 3 (Figure 1).

*3) It is unclear why cone arrestin increases in Arr3. This suggests that there may be off-target effects. Does ZF6-DB also bind to the Arr3 TSS region? ChIP data should be supplied to address off-target issues.*

The increase of cone arrestin in Arr3 data did not show a statistically significant difference compared to other experimental groups. However, as suggested we performed ChIP analysis amplifying the proximal promoter region of Arr3 after pull down. As shown in Figure 1—figure supplement 1 we did not observe pull-down enrichment of Arr3 promoter fraction after PCR amplification, thus confirming the apparent lack of physical binding of ZF-DB on the Arr3 proximal promoter. RNA-seq data also confirmed the lack of differential expression of the Arr3 transcript in the samples.

In addition, after adding a further sample in the silencing and replacement experiment (as stated in response to comment #2) the Arr3 levels do not appear to be different from the control.

*4) Figure 3 would benefit from the presentation of wild type mouse data. For example, eGFP, ZF6-KRAB, and ZF6-DB treatments of wild type mice to serve as a point of comparison and investigate the efficiency of ZF6-DB suppression of human and mouse rhodopsin respectively.*

As stated in response to comment #2, we added data on eGFP, ZF6-KRAB, and ZF6-DB treatments of wild type mice. As now shown in Figure 3 the ERG analysis did not show any detectable effects compared to untreated controls. This suggests that the absence of binding of ZF6-DB does not result in any detrimental effects (the mouse RHO TSS sequence diverges and lacks the ZF6-DB binding site).

*5) Regarding the repression-replacement experiment described in Figure 4, it is not clear why a different promoter (CMV) was used in the control ZF6-DB vector than that used for the ZF6-DB cassette (modified RHO promoter) in the in the experimental ZF6-DB* –

*RHO expression vector.*

We used the CMV that was used in the previous experiments (Figure 1) as an internal control for the experiment using the ZF6-DB-RHO expression vector (silencing and replacement). As shown in Figure 4, the expression of ZF6-DB driven by both the CMV and the mutant RHO promoters both generated similar porcine Rho repression (RT-PCR experiment) and ZF6-DB rod nuclei expression, as assessed by anti-HA immunostaining, respectively. Currently, we are testing a series of mutant promoter elements with different strengths at a fixed vector dose to determine the impact on expression of ZF6-DB and in turn on Rho repression levels with the final aim of improving the overall suppression of Rho on the whole retina. We believe that these experiments will serve better as part of a separate study.

*6) The level of repression of the RHO gene achieved in the repression-replacement experiment described in Figure 4 is moderate. It is likely that greater than 35% suppression of a mutant RHO allele would be needed to have a therapeutic effect for patients affected by retinal degeneration due to gain-of-function mutations in RHO. While the data reported in Figure 1 suggest that repression of RHO expression in transduced photoreceptor cells is closer to 90%, it is unclear if suppressing RHO expression in 35% of photoreceptor cells would be therapeutic. Additional experiments to optimize the dose response and evaluate the potential of this approach for therapeutic use are needed to support the claim that a ZF6-DB based approach has the potential to be used clinically.*

As stated in responses to comment #1 and #2 we agree that the 35% (which has now become 38%) of whole retinal transduction may not be therapeutically effective. Accordingly, as with comment 1, we changed the claim regarding the therapeutic effect and we stated in the Discussion: “Therefore, a further development of this proof-of-concept will include optimization of the design of the silencing and replacement double construct (tuning the strength of the promoter elements), vector selection and dose, and the surgical approach”.

*7) Clarity of presentation as indicated below by reviewers #2 and #3.*

Done accordingly.

[Editors' note: further revisions were requested prior to acceptance, as described below.]

*It looks substantially improved, but there is one aspect of the writing that needs work: various words or phrases that imply an absolute or extreme conclusion are over-stating the data. For example, in the Abstract is the phrase "complete gene silencing". The data do not support the use of the word "complete"* – *indeed, quantitative biological data only rarely support the use of that word. The best one can say of any observation of this type is that the signal is below the limit of detection. I suggest substituting the word "efficient" for "complete". Similarly, in the title of the legend to Figure 1 is the phrase "abolishes Rho expression". I suggest changing this to "dramatically reduces Rho expression". Please go through the manuscript carefully to make changes of this sort. My view is that your data speaks for itself, and that the manuscript should be carefully written so as to avoid over-stating the data.*

According to the suggestions, we went through the manuscript carefully to mitigate absolute conclusions that may not be supported by the data.

The point-by-point changes are as follow:

In the Abstract: “efficient” replaced “complete”.

In the section “Photoreceptor genomic binding of a 20 bp-long DNA sequence by a synthetic DNA-binding protein turns off Rhodopsin expression”:

“complete” was removed, resulting in the sentence that states: “showed loss of EGFP expression compared[…]”.

“complete” was removed, resulting in the sentence that states: “Immunofluorescence analysis showed depletion of Rho protein[…]”.

“strongly” was removed, resulting in the sentence that states: “These data suggest that, despite the presence of an active canonical repressor domain[…]”.

“exclusively” was removed, resulting in the sentence that states: “ZF6-KRAB generated *Rho* silencing by DNA binding[…]”.

“any” was substituted with “detectable”, resulting in the sentence that states: “ZF6-DB in C57Bl/6 wild type retina did not produce detectable functional detrimental effects[…]”.

“balance” was removed resulting in the sentence that states: “transcriptional repression of the porcine *Rho* (38%) and in concomitant replacement of the exogenous *hRHO* (68%)[…]”.

“full recovery” is now substituted with “complementation” resulting in the sentence that states: “Notably, *Gnat1* transcript and protein (data not shown) levels demonstrated complementation[…]”.

In the first sentence of the discussion we substituted “turn off” with “dramatically reduces”.

The following sentence: “The retina co-transduced with eGFP and ZF6-DB vector showed virtually a “somatic knock-out” of *Rho* expression (~85% decrease of Rho transcript levels)”, may appear an over-stating claim. However, we believe that “virtually” counterbalances the strong “somatic knock-out”, highlighting the impact of the finding.

In the title of the legend to Figure 1 the sentence "abolishes Rho expression" is in now changed with "dramatically reduces Rho expression".

We changed the sentence: “We propose a model in which the molecular features of the DNA (loop, twisting, bending, position, for instance) may contribute to the transcriptional interference observed” with: “We propose a model in which the molecular features of the DNA (loop, twisting, bending, position, for instance) may contribute to *Rhodopsin* transcriptional output”.

We added the following sentence that we believe completes the theoretic thinking in the Discussion:

“It follows that in terms of signaling, the information source, the DNA, generates an output signal the RNA and eventually a protein, whose final output functional activity is completed back by the DNA. Thus the information source, the DNA, becomes also integral part of the signaling (*Rhodopsin* transcriptional output)”.

Regarding the impact statement, I would favor the concise message of “turns off”: “Photoreceptor genomic binding of a 20 bp-long DNA sequence by a synthetic DNA-binding protein turns off Rhodopsin expression”.